# Geoheritage and Geoconservation: Some Remarks and Considerations

Eva Pescatore, Mario Bentivenga and Salvatore Ivo Giano *

Department of Science, University of Basilicata, Campus Macchia Romana Via Ateneo Lucano, 10,
IT85100 Potenza, Italy
* Correspondence: ivo.giano@unibas.it

**Abstract:** Topics related to geoheritage research, protection, and conservation, as well as the enhancement and dissemination of geoheritage knowledge, have experienced an important increase in interest regarding the perspectives of both research and management policies. In geoheritage and geodiversity management, geoconservation is a term that encompasses a series of actions dedicated to conservation, research on and the protection of geoheritage, and the enhancement as well as dissemination of knowledge in this area. Geoconservation is a kind of container, with several compartments dedicated to different aspects that identify geoheritage and geodiversity, including scientific, technical, administrative, didactical, and political aspects. These aspects are necessarily different according to (i) objects directly or indirectly involved in geoconservation actions; (ii) the area of application (protected and unprotected natural areas; emerged, submerged, or mixed areas; and urban, urbanized, and/or anthropized areas); (iii) final goals; and (iv) the final end users. This paper presents a schematization of geoconservation concepts and applications as expressed in the literature and as a result of personal experience in addressing issues related to geoheritage management.

**Keywords:** geoconservation; geoheritage; physical landscape; territorial planning; geoheritage management

## 1. Introduction and Aims

In recent decades, topics related to the study, protection, and conservation of geoheritage, as well as those related to the enhancement and dissemination of knowledge of the field, have experienced an important increase in interest in both academic research and management policies dedicated to these issues. Often, when talking about the natural world, biotic aspects (plants and animals) are considered endangered, at risk of extinction, and susceptible to many threats, while abiotic aspects (rocks and morphologies) are considered solid, robust, and abundant; therefore, they do not require special study or protection [1–3]. Environmental protection policies have been mainly aimed at the biotic aspects of natural environments; academic research only began to address the abiotic aspects of natural environments in the late 1990s [2,4–8]. The common perception of geological processes' "slowness" and of rocks' "robustness" can lead to erroneous interpretations regarding their interactions with the natural environment and their evolution in terms of intensity as well as over space and time. This perception can lead to the misconception that geological resources are inexhaustible and immutable, while many unique and unreproducible landscapes as well as outcrops have already been destroyed forever due to inappropriate management [1]. Another misconception is that the time required for certain natural phenomena to occur, or to reoccur, is so long as to lead one to underestimate, or even ignore, the phenomena themselves. In a society focused primarily on the "now", which often ignores history and has little interest in what might happen in the not-so-distant future, relating the geological dimension of natural systems' space–time evolution to the human experience is not simple. As pointed out by Dodic and Nir (2006) [9], "*Human beings*

*are limited to a lifetime that will allow them to see (with good health) the passage of three generations, not nearly the time needed to psychologically encompass the 4.5 billion years of earth history. Thus, the question remains as to how it might be possible to cognitively understand (and accept) the vastness of geological time and the events (both biological and geological) which have shaped our planet*". Some activities and actions aimed at geoheritage as well as geoconservation could be a link between these two aspects: conserving an object for present and future use, taking into account its possible evolution in the short, medium, and long term, and using it as a tool to make the concepts of geological time and landscape evolution time real and understandable. Planning geoconservation actions starts with the definition of the primary focus, represented by geosites and areas of geological interest. By the term "areas of geological interest", we mean sites or areas that, while not having a character of particularity or rarity in geological significance, are significant for didactic, tourist, cultural (the geodiversity site in Brilha, 2016 [10]), or territorial planning activities, and could represent an opportunity for local development. As such, they can be the subject of geoconservation actions, similar to geosites. For the purposes of this paper, geosites and areas of geological interest are considered together and are hereafter referred to as GSs.

Several papers are dedicated to geoconservation in natural protected areas. Geoheritage is not always confined to protected natural areas; geological objects with scientific/cultural value are often present in anthropized or urbanized areas. In these cases, hypothetically, an approach similar to that for natural protected areas could be used; in reality, however, the approach should be different because the actors involved are different, as may be the actions, needs, and final objectives. The aims of this paper are (i) to propose a procedure for addressing issues and actions for geoconservation purposes, and (ii) to focus on the role as well as potential of geoconservation issues and actions in various fields (scientific research, education, communication, and heritage management) and contexts (natural, natural protected, or anthropized areas). The model presented in this paper is the subject of practical applications; these will be discussed in future papers.

## 2. Background

Several concepts and definitions regarding geodiversity, geological heritage, geosites, geoheritage, and geoconservation are reported in the literature (e.g., [11–13]). Terms such as geoheritage, geodiversity, and geoconservation [14–19] are preferable to "geological heritage", "geological diversity", and "geological heritage conservation", since, in the collective imagination, these latter terms are more easily associated with the concept of solid rock than with a set of abiotic forms, materials, and processes (Sharples, 2002) [17]. Brocx and Semeniuk (2007) [20] define the following as belonging to geoheritage "*Globally, nationally, state-wide, to local features of geology, such as its igneous, metamorphic, sedimentary, stratigraphic, structural, geochemical, mineralogic, palaeontologic, geomorphic, pedologic, and hydrologic attributes, at all scales, that are intrinsically important sites, or culturally important sites, that offer information or insights into the formation or evolution of the Earth, or into the history of science, or that can be used for research, teaching, or reference*". Brilha (2016) [10] proposed restricting the use of the word geoheritage to elements with high scientific value, namely geosites for in situ elements and geoheritage elements for ex situ ones, and to restrict the use of the word geodiversity to elements with educational, aesthetic, and cultural value, in addition to scientific value, namely geodiversity sites for in situ elements and geodiversity elements for ex situ ones.

Geodiversity is a descriptive term [5,11,16,20–23] that includes (i) the different aspects of the world of geology, from sedimentological, volcanic, and climatic elements to the landscape and its change, to name a few; (ii) the wide range of phenomena and processes that create, or have created, landscapes, rocks, minerals, fossils, and soils, and that represent the basis for the presence of life on Earth [22]; and (iii) "*the link between people, landscapes and their culture through the interaction of biodiversity with soils, minerals, rocks, fossils, active processes and the built environment*" [22]. Geodiversity is indicative of the natural variety of all of Earth's geological, geomorphological, and pedological aspects, including their

associations, relationships, properties, interpretations, and systems [11]; therefore, geodiversity is present everywhere in the landscape, in rocks and even in building stones and buildings. The elements that make up geodiversity do not always have scientific importance; however, they can be important elements from a cultural, tourist, or community point of view [10], as well as playing a more or less relevant role in the definition of the physical landscape in which they fall and its evolution, depending on the scale of observation. Geodiversity supports biological habitat diversity; habitat diversity, as well as the number of species associated with habitats, is greater in areas with greater abiotic diversity, represented by an articulated topography, the presence of corridors, the soil types, and the prevailing geomorphological processes [21,24]. As with biodiversity, geodiversity is sensitive to environmental changes; similarly, since biodiversity is also dependent on geodiversity, the success of biodiversity conservation (bioconservation) requires integration with geoconservation [17]. Geodiversity has a fundamental value in maintaining vital ecosystems and thus ensuring biodiversity, as it supports the diversity and variability of habitats on different temporal and spatial scales [24].

The first to use the word geoconservation was Sharples (1993) [15]. He pointed out that geoconservation is important for "*The maintenance of natural earth processes (especially geomorphological, hydrological and soil processes) in order to ensure the sustainability of ecosystems (of which they are an integral part) as a whole. Earth systems and processes (encompassing landforms, soils and bedrock geology) exert a major control over, and interact with, the biotic communities which sit upon them*". Geoconservation, therefore, recognizes that the natural environment's non-living components are equally important for the conservation of the environment itself and its living components, and thus are equally in need of adequate management. Because geodiversity provides a variety of environments and environmental matrices that directly affect biodiversity, the degradation of a landscape, including its soil and water, will have a negative impact on species and biological communities living in or on them [17], including humans. The goals of geoconservation include the conservation of natural environments' geological aspects, the containment of the rate and extent of natural changes acting on natural environments [17], and the containment of the impact of anthropogenic activities, current and historical, on natural environments [25]. These considerations were summarized in [11] as follows: damage to or the partial or complete loss of an element of geodiversity; the interruption of natural processes and off-site impacts; loss of interest, visibility, or intervisibility; loss of accessibility; pollution; and visual impacts.

Geoconservation also includes the set of all legislative provisions, administrative tools, applied measures, and techniques of analysis, management, and evaluation, including recovery and redevelopment [26,27], as well as functional measures for the growth of the three main components of sustainable development: the environment, society, and the economy. Geoconservation is a process that begins with the awareness of the existence of geodiversity, followed by the assessment, enhancement, recognition of danger as well as risk, and protection of geoheritage through legislative acts with a holistic and/or integrated approach [28].

Henriques et al. (2011) [29] applied concepts and schemes related to the paleontological context (according to Fernández-López (2000) [30]) to the geoconservation context, distinguishing between basic geoconservation (the classification of Earth's geological heritage), applied geoconservation (the conservation of Earth's geological heritage), and technical applications of geoconservation (the valuation of Earth's geological heritage). Crofts et al. (2020) [3] outlined the hierarchy of the related concepts: "*Geodiversity is the totality of abiotic nature, of which some elements have significant value requiring conservation, termed Geoheritage, which is managed in Geosites, that are either formally protected areas or are conserved areas, under the generic label Geoconservation. The overriding purpose of Geoconservation in protected and conserved areas is to conserve geoheritage and geodiversity located in geosites*".

## 3. Geoheritage, Geodiversity, Physical Landscape, and Geoconservation

According to [10], geoheritage is a term reserved for objects with scientific value. Geodiversity is a broad term referring to areas of educational/tourist interest and to a landscape in general. Geoheritage and geodiversity are integral parts of the physical landscape (hereinafter referred to as PL) in which they occur. The evolution of a PL can significantly influence their evolution. Words such as "environment", "territory", and "landscape" are often used as synonyms, indicating what surrounds us in a generic way or in a more specific manner by adding the adjectives natural, uncontaminated, anthropic, industrial, rural, etc. The European Landscape Convention [31] is the first international treaty exclusively dedicated to the European landscape as a whole, taking into consideration natural, rural, urban, and peri-urban spaces, whether they are exceptional or ordinary, recognizing their relevant role in the quality of life of their inhabitants. The Convention, in encouraging and fostering European cooperation, aims to promote the protection, management, planning, improvement, and, where necessary, the creation of European landscapes. Several disciplines are involved in the study of the environment/territory/landscape, each of which mainly highlights its own disciplinary aspects in defining and analyzing general and specific characteristics. Sometimes, even within the same disciplinary context, the definitions may vary according to the area of expertise. For the purposes of this paper, the definitions contained in the ISPRA-CATAP Dynamic Glossary (https://www.isprambiente.gov.it/files/pubblicazioni/manuali-lineeguida/mlg-78.1-2012-glossario-dinamico.pdf) (accessed on 10 November 2022) and in Amadei et al. (2000) [32] are used. Furthermore, beyond the etymological definition, the generic meaning, and the disciplinary context of reference, a PL is characterized by (i) abiotic and biotic components in addition to (ii) anthropogenic components (Table 1). The latter, although included in both the biotic components (man is part of the biosphere) and abiotic components (anthropic structures), are treated separately as components with a strong, significant, and, in some cases, decisive "environmental" impact, in both a positive and negative sense. PL evolution is influenced by various factors: geology (the nature, distribution, and structural set-up of geological bodies); water (hydrography and water bodies); climate; geomorphic processes (the nature and speed of exogenous as well as endogenous processes); vegetation (the nature and distribution of spontaneous and/or cultivated species); fauna (the nature and distribution of wild and/or farmed species); and anthropic factors. The nature and speed of geomorphic processes can change over time, producing different shapes in different environmental conditions. Time represents an important factor, as what we currently observe is the result of various events that have occurred over time, overlapping, sometimes adding up, and sometimes obliterating the traces of the previous events. Landscapes are dynamic, in continuous evolution, with times and modalities that, brought back to the human scale of space–time perception, may seem ephemeral and somewhat irrelevant. The human perception of time and the timing of changes, as well as of landscape evolutionary dynamics, at any scale, can represent a significant obstacle in the management and planning of natural heritage.

Natural changes in geodiversity related to PL evolution may be rapid (such as changes related to floods, landslides, volcanic eruptions, earthquakes, and so on), slow, or extremely slow. The reaction of biodiversity to geodiversity changes may be very fast; without optimal biological parameters (also supported by abiotic geological contexts), biological species may migrate or become extinct. In contrast, in the presence of optimal biological parameters created as an involuntary consequence of territorial management interventions, alien or invasive species can easily and quickly adapt and spread, with consequent harm to endemic species.

Among the abiotic components, GSs need a dual approach, considering them as both single elements and as a function of the PL in which they exist. A multiscale approach is fundamental, as the dimensional scales of GSs can vary from the crystal scale to the regional one [2,20]. Additionally, consideration of the dimensional, temporal, and thematic scales should be taken into account. Figure 1 shows a schematic representation of different scales, extrapolated from Summerfield (1991) [33] as well as Brocx and Semeniuk (2007) [20]; the

reference scale could be an observation, analysis, study, or restitution scale, depending on both the studied object and on the final objective. Note that the following text refers to geoheritage; it can also be considered as referring to bioheritage, biosites, archeoheritage, archaeosites, and other sites of interest, with the appropriate amendments and changes.

**Table 1.** Physical landscape components and critical issues.

| PHYSICAL LANDSCAPE (PL) | | | | | | | |
|---|---|---|---|---|---|---|---|
| | Component | Abiotic | Geo | General frameworks | Geological, geographic-physical, geomorphological, paleontological, volcanological, mineralogical, climate, hydrogeological, geosites and sites of geological interest (GS), etc | | |
| | | | | | Geosites and sites of geological interest * (GS) | | |
| | | | No-geo | Anthropogenic structures | Agricultural | | |
| | | | | | Residential | | |
| | | | | | Historical | | |
| | | | | | Industrial | | |
| | | | | | Tourist | | |
| | | | | | … … | | |
| | | Biotic | Bio | Flora | No Natural ** | | Intensive |
| | | | | | | | No intensive |
| | | | | | Natural | | Alien |
| | | | | | | | Endemic |
| | | | | Fauna | Natural | Wild | |
| | | | | | | Farming | Widespread |
| | | | | | | | Intensive |
| | | | | Anthropic *** | | | |
| | Critical issues | | | Short, medium and long time evolution and vulnerability, depending on: | | components changes | |
| | | | | | | external pressure | |

* Although they are part of the Physical Landscape's Abiotic_Geo Component, GS are treated separately as basic elements for geoconservation interventions; ** GMO. *** Although it is part of the Physical Landscape's Biotic_Bio_Fauna Component, it is treated separately as a component with a high environmental impact and change power.

Geoconservation actions cannot be limited to the conservation of GSs; they must also be extended to the PL in which they are included, which can affect their spatial/temporal evolution, directly or indirectly. Managing geoconservation related to individual objects, without evaluating the natural system in which they are included, does not take into account the fact that individual elements can evolve/degrade as a result of events happening in another part of the system [15]. Considering the individual elements without their contextualization on a larger scale could lead to erroneous assessments of both the present situations as well as their evolutionary time projections, and therefore to ineffective or pejorative interventions. Furthermore, a PL may, and in some cases must, be subject to geoconservation actions even if it lacks a GS within it; one example of this is river environments, where geodiversity, also understood as morphological diversity, represents a key point for both geo- and bioconservation actions.

| Summerfield M.A. (1991) | | Present paper | | | | | | | | | |
|---|---|---|---|---|---|---|---|---|---|---|---|
| **Scale landforms** | | | **Scale Analysis / Interest** | | | | | | **Scale Restitution Action (Planning of)** | | |
| Megascale | > 1,000.000.km² | *Megascale* | Global | | | | | > 1,000.000.km² | Big | > 100,000 | *Indicative reference scale values 1: X* |
| Macroscale | 100-1,000.000 km² | *Macroscale* | Geodynamic | | | | | 500 -1,000.000 km² | Large | 100,000 50,000 | |
| | | | Regional | | | | | 100-500 km² | | | |
| Mesoscale | 0.25-100 km² | *Brocx & Semeniuk, 2007* | Regional | 100 km² | *Mesoscale* | *Outcrop scale* | Local scale | 10 - 100 km² | Medium | 10,000 5,000 | |
| | | | Large | 10 km² | | | Large | 1 -10 km² | | | |
| | | | Medium | 1 km² | | | Medium | 100 m²-1km² | | | |
| Microscale | < 0.25 km² | | Small | 10 -100 m² | | | Small | 10 m² - 100 m² | Small | ≤2,000 | |
| | | | Fine | 1mx1m | | | Strata scale | 1 -10 m² | | | |
| | | | Very fine | 1 mm x 1 mm | | *Microscale* | | 1mm² | | | |

**Figure 1.** Schematic representation of different scales [20,33].

Actions related to geoconservation represent an opportunity for knowledge advancement and for multidisciplinary scientific as well as technical collaboration, involving figures from different backgrounds and an opportunity for (a) increasing (i) local knowledge of one's own territory and sociocultural roots; (ii) territorial enhancement, including locally, nationally, and internationally; (iii) the citation of natural and environmental science basic knowledge, with particular regard to issues related to climate change, pollution, risk, etc.; and (iv) social as well as economic development, (b) highlighting (i) environmental and territorial issues in addition to (ii) topics linked to climate change and pollution.

In general, geoconservation actions can be both material (acting on an object) and virtual. Among the latter are the images that can testify to the evolution of an object over time and virtual reality, which allows us to observe an object outside of the context in which it is located or to reconstruct its past evolution and hypothesize about its future development. Virtual reality and augmented reality represent important tools for geoconservation actions, as they allow us to create usable objects and reconstruct natural scenarios that otherwise could be difficult to manipulate or reach by end users with limited motor or sensory capacity. Geoconservation also means making an object usable without linguistic (in terms of the content and language used) or physical barriers, when possible, or choosing and suitably preparing sites/paths/contents that allow this.

By modifying what was proposed by Henriques et al. (2011) [29] and taking into account schemes proposed by Brilha (2016) [10], it is possible to distinguish different levels of analysis, research, and intervention in the field of geoconservation, closely related to each other, as well as different goals, languages (lexicon, syntax, and morphology) used in text, and end users (Figure 2).

It is vital that the language used must be compatible with the end goals and appropriate for those who will be the end users. As mentioned above, GSs are considered as both an integral part of a PL and analyzable as individual elements where a PL contains one or more GS; if there are no GSs, the PL, when representing an object subjected to geoconservation, can be analyzed by using the same approach, as described below.

- **Basic Geoconservation**—GS and PL definition and characterization, analyzed within the environmental, cultural, and socioeconomic contexts in which they are included; specific studies on potential vulnerability in the short, medium, and long term.
- **Applied and Popularizing Geoconservation**—GS and PL evaluation and classification; geoheritage management database production; geoconservation general action

guidelines; territorial planning guidelines; geoheritage, scientific, and territorial knowledge dissemination; environmental issue dissemination; and geological tourism.

- **Technical Geoconservation**—The production of material and proposals aimed at supporting geoconservation actions and interventions, distinguishing short-, medium-, and long-term actions, with the aim of defining guidelines that can be re-proposed in different contexts; the definition and planning of short-, medium-, and long-term geoconservation actions; the definition and planning of small, medium, and large spatial-scale geoconservation interventions; evaluation, proposal, and validation of the possible use of mixed/integrated consolidation/stabilization and geoconservation techniques; and support activities for geoconservation actions carried out in non-geo-contexts.

| GEOCONSERVATION | | |
|---|---|---|
| **Basic Geoconservation** | **Applied and Popularizing Geoconservation** | **Technical Geoconservation** |
| **Actions** Production of materials and proposals aimed at: characterization of geo-heritage and of landscape-environmental, cultural and socio-economic context in which geo-heritage is inserted (geo and non-geo aspects), also including specific studies on their potential vulnerability in the short, medium and long term | Production of materials and proposals aimed at: geoheritage evaluation and classification; geoconservation actions guidelines; geoheritage database production; geoheritage and scientific and territorial knowledge dissemination; environmental issues dissemination | Production of materials and proposals aimed at: geoconservation actions and territorial planning, distinguishing short, medium and long term actions. Support activities to geoconservation actions carried on in non geo contexts |
| **Goals** Scientific papers | Scientific papers Managerial materials General/generic planning guidelines Didactical materials Promotional material | Scientific papers General/generic and/or specific intervention guidelines |
| **Language** Scientific | Scientific Technic Popular | Scientific Technic |
| **Main end-users** Academics | Academics Public/Private landscape mangers Educational institutions Tourist and territorial promotion institutions | Academics Public/Private landscape mangers |

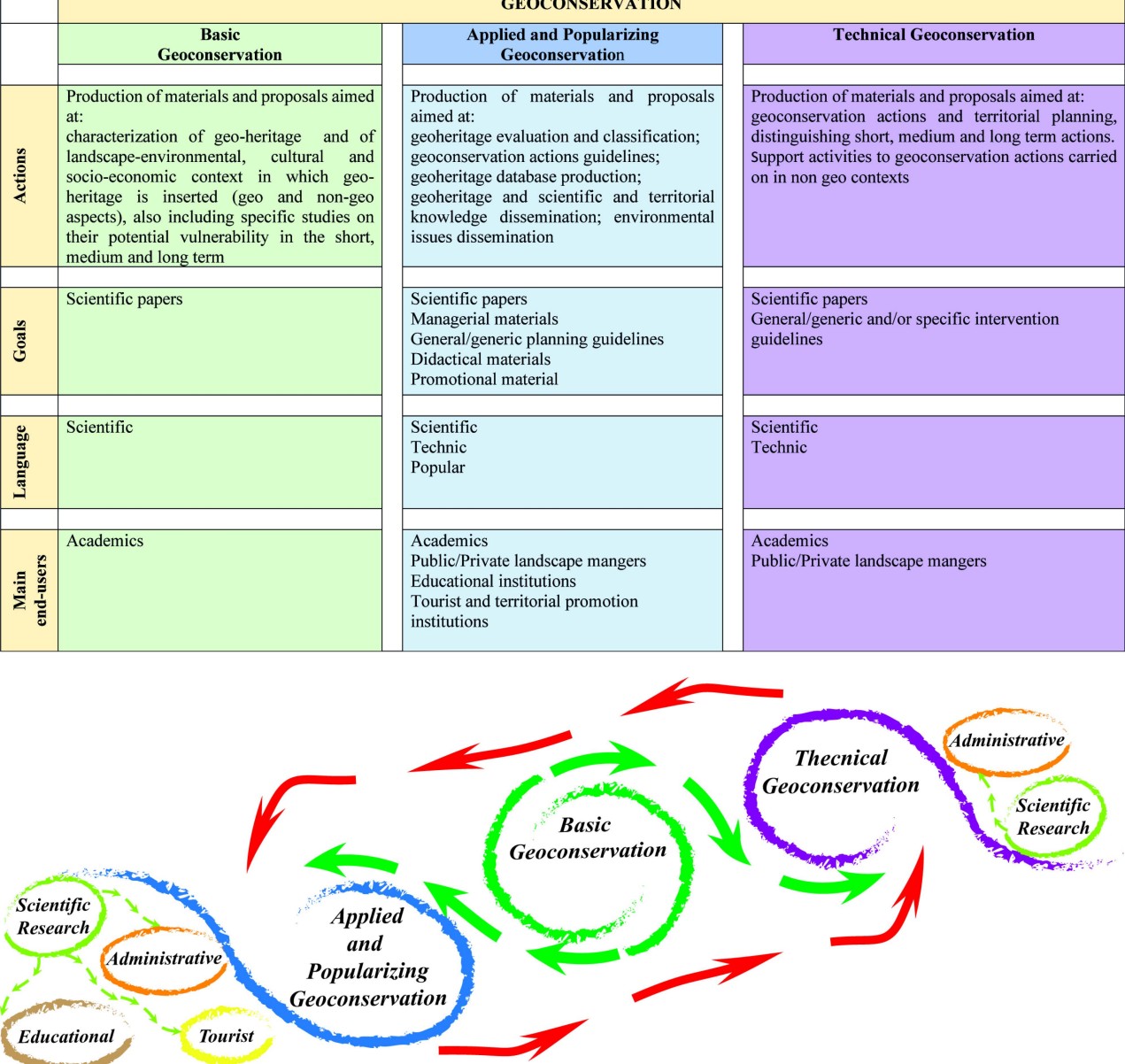

**Figure 2.** Sketch of a table (**up**) and flow diagram (**down**) showing the proposed schematization. See the text for further explanation.

Step 0 is represented by the definition and delimitation of the area to be studied as well as by the following: the distinction of the type, or types, of PL present; the definition of the number and type of geosites present; the definition of the number and type of areas of geological interest present. If no GSs are present, only the PL will be defined (Figure 3).

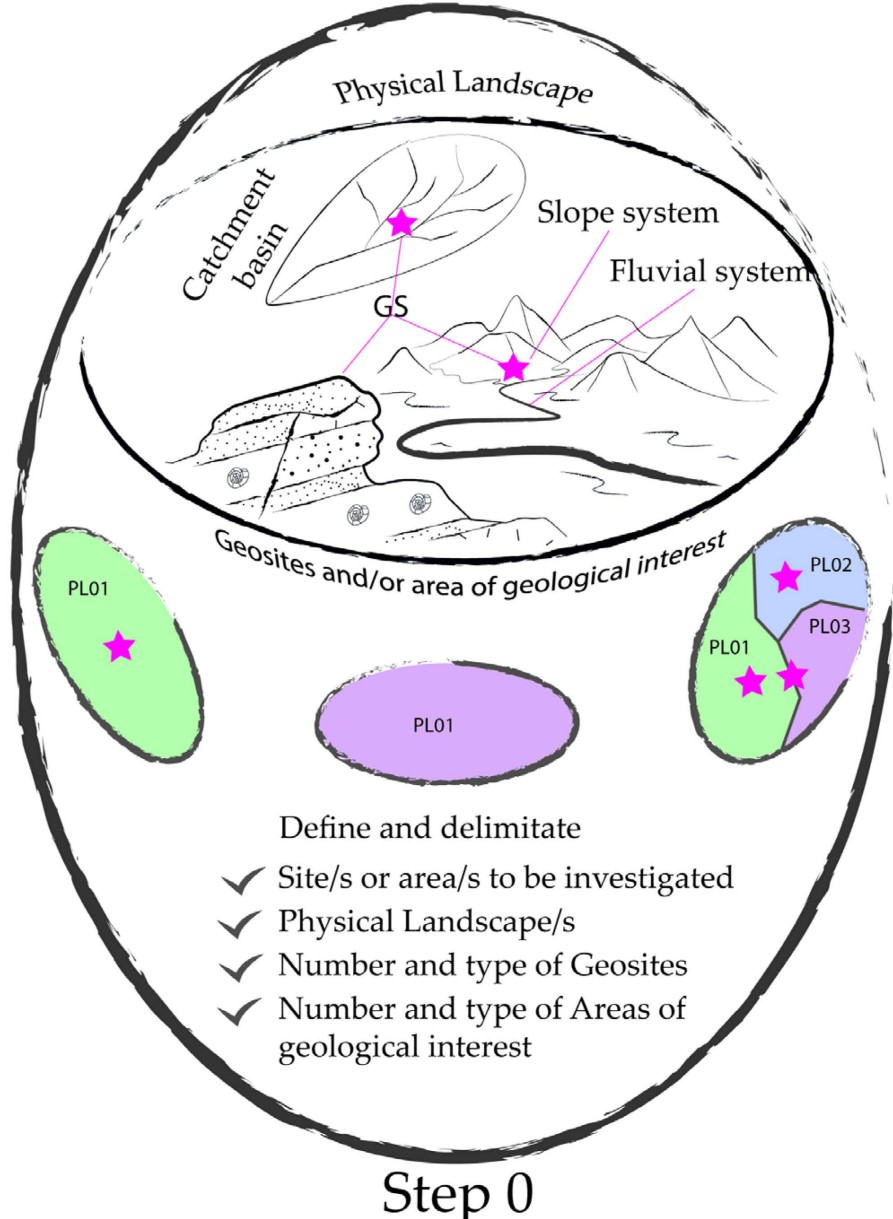

**Figure 3.** Step 0—Definition of the area that will be studied: define the type, or types, of PL present; define the number and type of present geosites; and define the number and types of areas of present geological interest. Red stars represent GS.

## 4. Basic Geoconservation

Basic Geoconservation is the basic knowledge, characterization, and study of GSs and PLs, as well as of the territorial, natural, and sociocultural contexts in which they are located, highlighting geo-contexts [12,34–37] and non-geo-contexts [38–40]. The study scale may vary from a micro- to a mega-scale, depending on the type of study, the object of the study, and the purpose of the study. Actions in this context are related to a PL's definition and characterization, including the components that define it and have an impact on its evolution in terms of their current state, their possible evolution over time, and their criticality as well as vulnerability. In the context of a PL's abiotic geo-components,

GSs must be analyzed and characterized (i) as single entities, at an adequate scale of analysis in terms of the nature and types of GSs; (ii) according to the physiographic unit, represented by the hydrographic basin (one or more) in which they are included, as defined by Fryirs and Brierley (2013) [40]; and (iii) according to the PL in which they are included. Thus, different scales of analysis and characterization are provided: a detailed scale (in situ scale), a hydrographic basin scale, and an overall scale (as an element within the PL). The main purposes of keeping pace with technological and knowledge advances are to (i) ensure the updating of scientific knowledge; (ii) prevent the outdating of scientific knowledge; (iii) promote scientific knowledge exchange; and (iv) monitor scientific knowledge advancement. The main goal is represented by the authoring of scientific papers, the best way to achieve the aforementioned purposes (Table 2).

**Table 2.** Step 1. Basic geoconservation: main actions and purposes.

| Basic Geoconservation | | Physical Landscape | | | | | | |
|---|---|---|---|---|---|---|---|---|
| ✓ General framework characterization and definition<br>✓ Bibliographic background<br>✓ Characterization and definition of the main geo and no-geofactors that have affected and affect, at present day: sites; structures; bio-components<br>✓ Evolution, criticality, and vulnerability over time, related to the main factors, geo- and non-geo-, that have affected, are acting currently, and will affect sites; structures; and bio-components<br>✓ Criticality, impact, and evolution over time, with respect to any anthropic and touristic component of sites; structures; and bio-components | **Component** | **Abiotic** | Geo | | Geo-contexts | As a physical landscape's main components | | |
| | | | | | **Geosites** and/or sites of geological interest | | | |
| | | | Non-geo | Anthropogenic structures | As essential parts of the physical landscape in which they are included | | | |
| | | **Biotic** | Bio | Flora | | | | |
| | | | | Fauna | | | | |
| | | | | Anthropic | | | | |
| | **Critical Issues** | Component changes | Abiotic | Geo | i.e., climate change, seismic risk, volcanic risk, etc. | | | |
| | | | | Non-geo | As parts of the physical landscape | | | |
| | | | Biotic | | As geomorphic agents | | | |
| | | External pressure | Anthropic | | | | | |
| **Main Purposes** | | | | | | | | |
| Ensure | Prevent | | Increase | | | Monitor | | |
| Scientific knowledge | | | | | | | | |
| Updating | Ageing | | Exchange | | | Advancement | | |
| according to knowledge and technological advancement progresses | | | | | | | | |

This is Step 1 in the approach to geoconservation actions; it has a strong role in subsequent actions to evaluate and classify GSs and PLs as well as identify the most appropriate strategies to protect, improve, and monitor geoheritage (Figure 4).

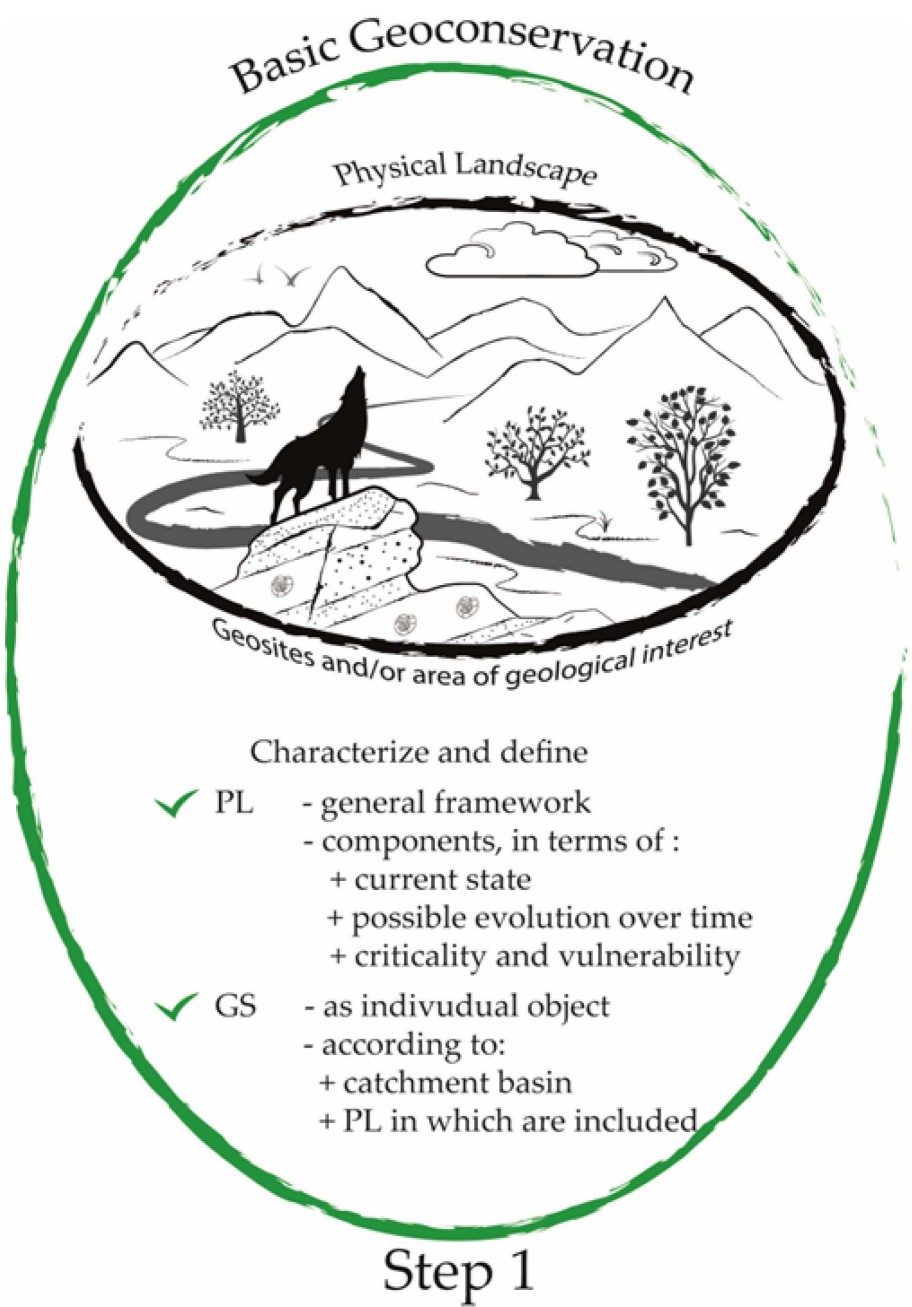

**Figure 4.** Step 1: Basic geoconservation main action. Silhouette of trees designed by Freepik.

## 5. Applied and Popularizing Geoconservation

Actions included in this context involve different actors and are dedicated to a wide audience, which includes universities, educational institutions of all types, territorial and tourist promotional institutions, tourists in general, and local administrations. Simplifying, in this context we can group the end users into four macro-areas: Scientific Research; Administrative (local, provincial, regional, and national); Educational; and Tourist. Each of these macro-areas is characterized by different main requirements, approaches, final purposes, goals, and languages (in terms of concepts and content, as well as the use of appropriate vocabulary). In this preliminary schematization, we opt to include both applied and popularizing actions in the same setting. This is because the actions envisaged for the Scientific Research macro-area represent the starting point for the actions envisaged for the other macro-areas, as we will explain below. We do not exclude, for the future, the separate development of the two categories of Applied and Popularizing Geconservation (Figure 5).

| | Applied and Popularizing Geoconservation | | | |
|---|---|---|---|---|
| | **Applied** | | **Popularizing** | |
| **Main Requirements** | **Scientific Research** | **Administrative** | **Educational** | **Tourist** |
| | • Site/area | *Actions* • Assessment of | • Contents | • Contents |
| | Evaluation \| Classification | ✓ Feasibility | In situ \| Extra situ | In situ \| Extra situ |
| | According to its | ✓ Criticality | ✓ By age group | ✓ By tourist type |
| | ✓ Rarity | ✓ Safety | ✓ Interdisciplinary | ✓ Interdisciplinary |
| | ✓ Value \| Scientific | *Impact* ✓ Economic | • Site | • Site |
| | Cultural | ✓ Employment | ✓ Accessibility | ✓ Accessibility |
| | Aesthetic | ✓ Social | ✓ Safety | ✓ Safety |
| | Didactical | ✓ Environmental | ✓ Facilities | Facilities |
| | ✓ Integrity/conservation state | • Keywords | • Keywords | • Keywords |
| | ✓ Interdisciplinarity | | | |
| | ✓ Sociocultural link | | | |
| | • Keywords | | | |
| | • Acknowledgement in the | | | |
| | ✓ Scientific | | | |
| | ✓ Institutional *Sphere* | | | |
| | ✓ Social | | | |
| **Main Purposes** | See basic geoconservation | Territorial planning, management, and protection | Education, inclusion, and integration | Territorial promotion |
| **Goals** | • Scientific papers<br>• Support papers and materials to | • Territorial protection<br>• Territorial community/ies participation<br>• Local and territorial economic development | Scientific and territorial education | Sustainable tourism |
| **Language** | Scientific<br><br>Technical<br><br>Popular | Technical | Popular | Popular |

**Figure 5.** Step 2: Applied and popularizing geoconservation macro-areas, main requirements, and purposes. The red lines show correlations between the macro-area of scientific research and the others. See the text in the table for further explanation.

In this context, actions in the macro-area of Scientific Research represent a second step in geoconservation activities. Regarding the other macro-areas, the acquisition of materials produced by the Scientific Research macro-area is the first step after establishing the area of intervention (Step 0, Administrative), the theme to be developed (Step 0, Educational), or the type of tourism (Step 0, Tourism). Close collaboration between the macro-area of Scientific Research and the others is desirable in order to produce materials that meet the main requirements of the macro-areas in terms of the content, keywords, and languages adopted (Figure 6).

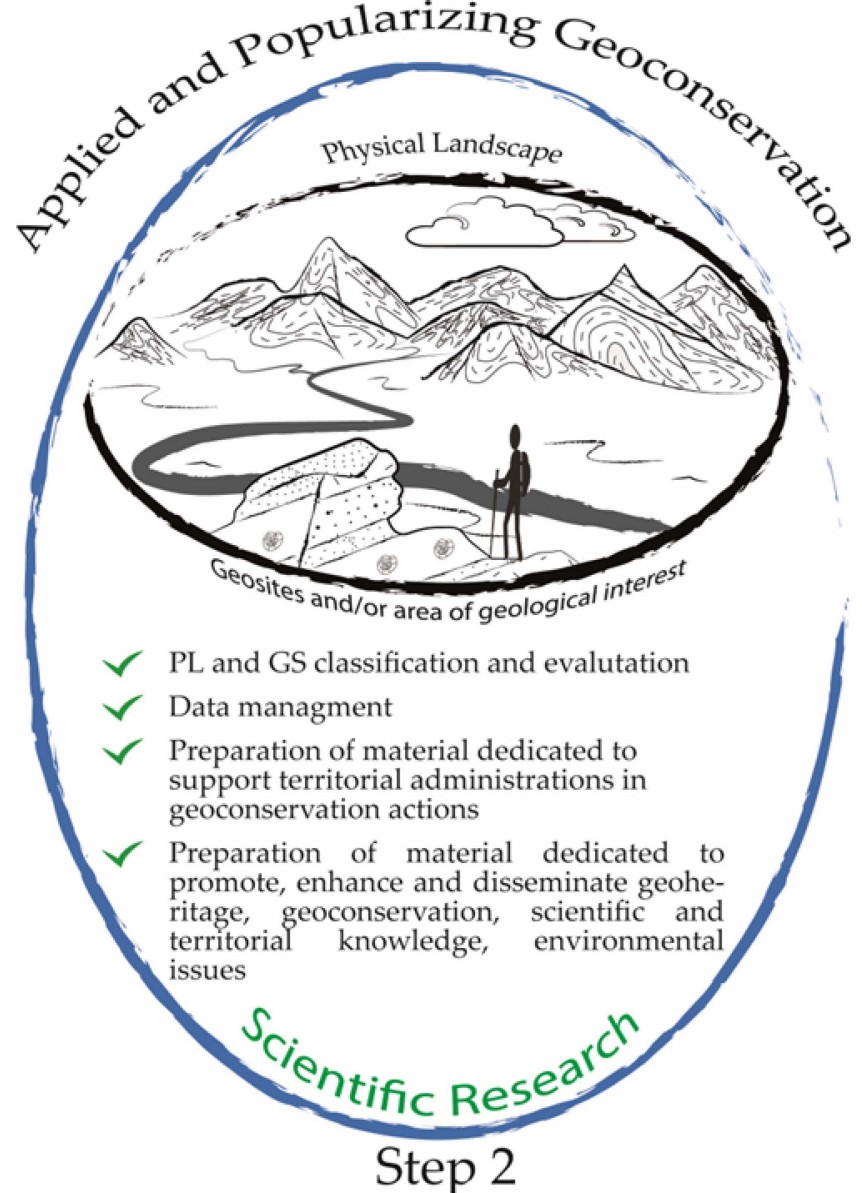

**Figure 6.** Step 2: Applied and Popularizing Geoconservation: Scientific Research macro-area main actions.

In the Scientific Research macro-area, actions are dedicated to (i) GS and PL evaluation and classification via the use of different methodologies as well as techniques of identification, cataloguing, and evaluation; (ii) the elaboration of relational databases dedicated to the dissemination of geoheritage and related or correlated issues; (iii) the preparation of materials dedicated to supporting territorial administrations in geoconservation actions; and (iv) the preparation of materials dedicated to promoting, enhancing, and disseminating geoheritage, geoconservation, scientific as well as territorial knowledge, and environmental issues. The main purposes are the same as those in Basic Geoconservation. The main goals are represented by the publishing of scientific papers and support materials dedicated to (i) territorial management, planning, and protection (the main purposes of the Administrative macro-area); (ii) scientific and territorial education (the goal of the Educational macro-area); and (iii) multidisciplinary tourism and sustainable tourism (the goal of the Tourist macro-area). The language used is scientific, with regard to scientific papers; technical, with regard to materials for use by territorial authorities; and popular, with regard to materials for educational and tourist use. The latter should be organized according to age

group (Educational) and type of tourist (Tourist). The production of materials useful for the scientific dissemination, at different levels of detail, of dedicated services for geological tourism as well as technical specific knowledge allows for the establishment of strong links between geo-topics within society (education and scientific dissemination, nature conservation, territorial planning, geological tourism or geotourism, etc.). Geoconservation, being focused on the management of geological elements with scientific, educational, tourism, or cultural value, allows us to link awareness and social responsibility to a more conscientious use of resources and heritage. This also includes actions to encourage and support sustainable tourism, as defined by Italy in the Strategic Plan for Tourism 2017–2022 (PST) (MIBACT, https://www.ministeroturismo.gov.it/il-strategic-planoftourism/; accessed on 27 November 2022), via the provision of dedicated material. As pointed out by Burek and Prosser (2008) [41], geoconservation can be truly effective only with the presence of local buy-in, which in turn is effective if local action is taken.

## 6. Technical Geoconservation

In this context, actions are correlated with and consequent to those of the Scientific Research macro-area, and represent a third step in geoconservation activities (Table 3). Actions included in this context are dedicated to the production of materials useful for GS and PL management, protection, and enhancement action plans, as well as geoconservation activities, highlighting the scientific–technical elements related to them.

**Table 3.** Step 3. Technical geoconservation: main actions and purposes. See the below text for details.

| Technical Geoconservation | ✓ Production of materials and proposals aimed at: |
| | ○ Geoconservation actions; |
| | ○ Territorial planning. |
| | ✓ Support to the definition and planning of: |
| | ○ Short-, medium-, and long-term actions; |
| | ○ Small, medium, large dimensional-scale interventions. |
| | ✓ Evaluation of geoconservation actions effects on the physical environment and its components; |
| | ✓ Evaluation, proposal, and validation of the possible use of mixed/integrated consolidation/stabilization and geoconservation techniques. |

| | | **Main Purposes** | | | |
| --- | --- | --- | --- | --- | --- |
| | | Ensure | Prevent | Promote | Monitor |
| | | At an appropriate time/dimensional scale | | | |
| Geosites and/or sites of geological interest | Site | ✓ Integrity<br>✓ Equilibrium<br>✓ Visibility<br>✓ Accessibility<br>✓ Safety | ✓ Damage<br>✓ Increase in and acceleration of natural morpho-evolutionary processes | ✓ Geodiversity<br>✓ Use of environmentally friendly and sustainable technologies as well as techniques | ✓ Evolution over time |
| Physical landscape | System | | | | |

This aspect is aimed at identifying the best practices to be applied for the purposes of geoconservation, differentiated according to the object and to the context in which the object is included. While the latter will be discussed below, some aspects will be treated in a general way in the following text. The proposed best practices will be functions of the PL type (e.g., mountain, hill, epigeal or hypogeal, and emerged or submerged), the characteristics of its components, the type of GS present, and the techniques as well as technologies available. Geoconservation interventions aimed at ensuring the conservation and use of sites of geological–naturalistic interest cannot fail to take into account the general context in which they are inserted when requiring stabilization, consolidation, and safety interventions. Thus, it is advisable to make a distinction between reinforcement/stabilization interventions and preservation/geoconservation interventions; the latter can be associated with the former (priorities for the purposes of the safety of places used or intended for

use by humans) if and when the site allows their simultaneous realization; otherwise, the priority and prevailing need of the anthropic component can overlap with the natural one (biotic and/or abiotic). In any case, the proposed interventions cannot be separated from a careful analysis of the evolutionary dynamics and the result/benefit ratio from naturalistic and anthropocentric points of view. In this context, the actions included are aimed at (i) defining guidelines on the technical aspects of geoconservation actions, both general and reproducible in different contexts as well as specific to a particular context, and (ii) evaluating, proposing, and validating the possible use of mixed/integrated consolidation/stabilization and geoconservation techniques (this topic will be addressed and explained later in the text) (Figure 7). The time scale of the actions, as a dimensional scale of the interventions, varies according to the subject and the objective. For GSs, the interventions may be on a small or medium scale or in areas of limited extension, taking into account the need for object conservation and the safety conditions in cases of touristic sites or areas.

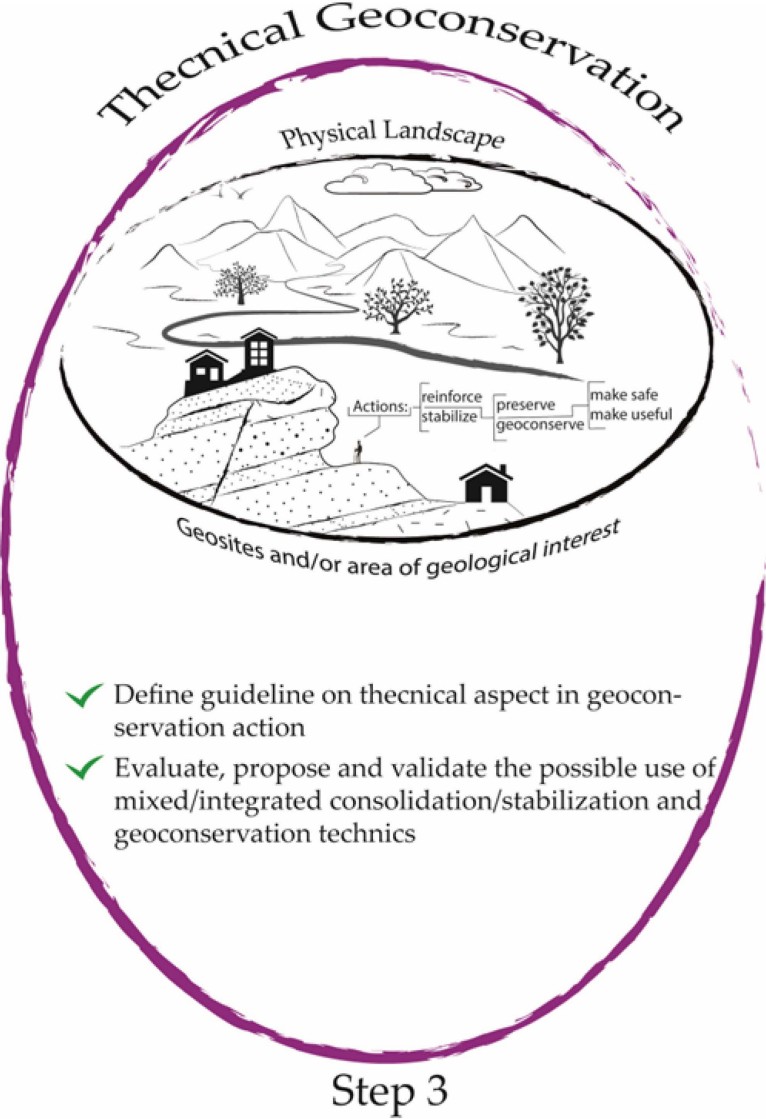

**Figure 7.** Step 3: Technical geoconservation: main actions. Silhouette of trees and buildings designed by Freepik.

With regard to PLs, interventions (medium- or large-scale) must take into account all of the components of a landscape itself and their criticalities, such as (i) the water sheet washing regimentation, whether in the case of a natural, urbanized, or anthropized landscape; (ii) the maintenance/creation/restoration of paths/infrastructures for both tourist and service uses; (iii) the conservation/restoration/creation of geodiversity (in addition to the morphological diversity along a watercourse, for example); (iv) slope stabilization; and v) safety requirements. The main aims with regard to GS site actions/interventions and to PL system actions/interventions, to be carried out at an appropriate time/dimensional scale, are (a) to ensure (i) the site integrity as well as equilibrium of abiotic and, if present, biotic components of naturalistic importance; for the latter, if not in balance, to evaluate the actions to restore balance; (ii) system equilibrium, and therefore the balance of abiotic and biotic components; (iii) visibility; and (iv) accessibility as well as safety (it makes little sense to secure a GS when the routes to access it are not secure); (b) to prevent (i) site damage and (ii) site/system increase in/acceleration of natural morpho-evolutionary processes as far as possible, or at least to significantly slow down these processes; (c) to promote (i) geodiversity, also referred to as morphological diversity, fundamental for biodiversity subsistence, and (ii) the use of environmentally friendly as well as sustainable technologies and techniques; and (d) to monitor the evolution over time of both the site and the system and, therefore, their abiotic and biotic components.

Actions involved in Technical Geoconservation could be (i) propositional, suggesting the best actions and techniques compatible with PLs, and ii) supportive, though the supervision of interventions as well as the monitoring and evaluation of their effects. Planned interventions could be (i) conservative, in the sense of preserving an existing object; (ii) regenerative, in the sense of restoring, improving, and thus preserving something that is subjected to natural decay processes; and (iii) creative, in the sense of recreating something that is currently lost but that, properly regenerated, will have a positive impact on the landscape.

Great steps have been taken in the fields of applied geology as well as geotechnics and engineering, from both technical and technological points of view, especially the use of naturalistic engineering techniques, which are preferred if and where applicable.

## 7. Remarks on the Proposed Guidance

The proposed approach emphasizes the manifold roles of scientific research in the field of geoconservation: (i) a strictly scientific research role, dedicated to deepening all aspects related to geoconservation actions, starting from the definition of objects and ending in their effective geoconservation; (ii) a role dedicated to the technological and managerial in-depth study of heritage management; (iii) a role dedicated to dissemination, fundamental to ensuring territorial knowledge and environmental respect, as well as the correct use of heritage, including geo- and non-geo-; and (iv) a role dedicated to the necessary support in the planning as well as implementation of geoconservation actions, which is the responsibility of those who know a landscape and its space–time dynamics. Roles that can "conflict" with other professional profiles represent an opportunity for growth in terms of cultural richness for those who act and awareness for those who receive. In Italy, examples of these collaborations in dissemination are represented by the activities that professional associations (https://www.cngeologi.it/t/a-scuola-con-il-geologo-iii-edizione/; accessed on 15 October 2022), national institutions (https://www.isprambiente.gov.it/it/attivita/formeducambiente/educazione-ambientale/programma-di-iniziative-per-le-scuole; accessed on 15 October 2022), and academic spin-offs are carrying out to bring the population closer to territorial and environmental issues.

Figure 8 highlights the central role that the Scientific Research macro-area plays in the study and production of materials that can be used in other contexts.

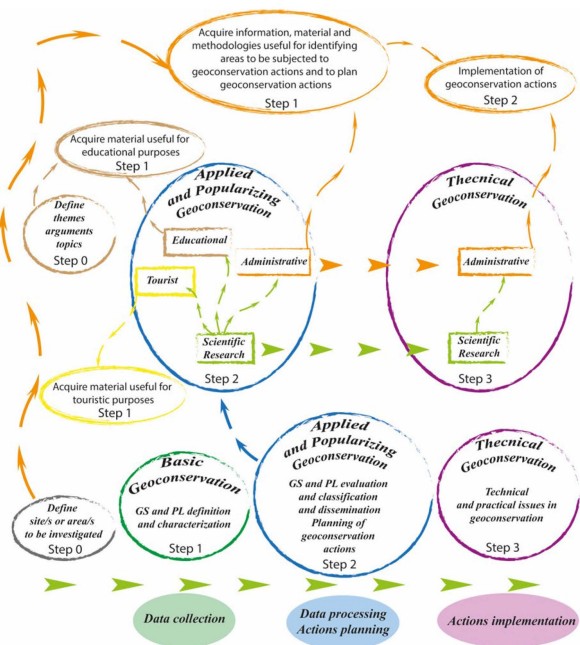

**Figure 8.** Schematic representation of the role of Scientific Research in the study and production of materials useful for other context. Large arrows refer to the step-by-step flow: green arrows are related to the Scientific Research macro-area actions; orange arrows are related to the Administrative macro-area actions; yellow arrows are related to the Tourist macro-area actions; and brown arrows are related to Educational macro-area actions.

## 8. Geoconservation and Areas of Application

Manuals [3,42], volumes [11,41–45], and copious articles in sector-specific as well as multidisciplinary journals are dedicated to issues related to geoconservation. UNESCO sites (World Heritage, https://whc.unesco.org/; Global Geoparks, http://www.unesco.org/new/en/natural-sciences/environment/earth-sciences/unesco-global-geoparks/; accessed on 20 November 2022) and IAG sites (http://www.geomorph.org/; accessed on 20 November 2022) are international reference sites; ProGeo sites (http://www.progeo.ngo/index.html; accessed on 20 November 2022) are the European reference sites; and the ISPRA website (https://www.isprambiente.gov.it/it; accessed on 20 November 2022) is the Italian reference site. Geoconservation interventions are necessarily different according to the territorial context in which they must be carried out; furthermore, the "human factor", intended as a geomorphic agent, must be taken into account. Several papers are devoted to anthropic activities that have a direct role in landscape modification [46–48], including both agricultural and pastoral activities as well as construction for civil, industrial, or defensive purposes and mining activities. All of these anthropic activities cause substantial landscape modifications, which can condition their stability and evolution; even a "temporary anthropic presence" (i.e., the presence of tourists can have a significant landscape impact. The anthropic component represents an essential part of a PL and therefore must be taken into consideration in the planning phases of geoconservation actions.

Geoconservation, in terms of studies, techniques, and implementation, can be considered simpler in natural and protected areas, unlike urban and/or urbanized and/or anthropized areas where the anthropic factor can enter as an element of complexity. With the words "urban area", we refer to a context in which the urban/constructive/infrastructural aspect is dominant over the territorial context, with an urban fabric that "covers" the entire territory (metropolitan cities, cities); by "urbanized area", we mean a context in which the urban/constructive/infrastructural aspect is less "covering" (cities, villages, and rural settlements) and sometimes more integrated into the landscape. In these contexts, where geoheritage can be represented by surface elements (outcrops and rock cliffs with significant geological value) or underground elements (natural or man-made cavities), talking about

geoconservation might seem inappropriate or in conflict with the priority of the human conservation aspect; in practice, in these contexts geoconservation can assume a dual role, one related to geoheritage conservation and one related to the application of mixed techniques for stabilization/geoconservation. The term "anthropized area" refers to those situations in which anthropic interventions have substantially changed a landscape, for better or for worse, as in agricultural landscapes, mining landscapes, or rectified rivers; the presence of tourist facilities also contributes to creating a context of this type. Shifting back to the anthropic factor as an element of complexity, this is understood in the sense that the needs and requirements related to the anthropic sphere may conflict (which is predominantly the case) with those of the natural sphere, including biotic and abiotic elements. Thus, it is appropriate to make a distinction between geoconservation in natural and protected areas as well as in urban and/or urbanized and/or anthropized areas, because the issues related to these areas are necessarily different, not regarding the definition and characterization of geoheritage but in terms of the determination of geoconservation issues, methods, and best practices.

Starting from the concept that geoconservation interventions can concern both GSs and PLs, Basic Geoconservation actions do not depend on the area of application, similar to those in Applied and Popularizing Geoconservation under the Scientific Research macro-area devoted to general themes; however, the constituents, participants, and contributing elements in the evolution of PLs and GSs must be defined, characterized, evaluated, classified, and prepared for use in application. Unlike actions envisaged in the Scientific Research macro-area devoted to planning geoconservation, actions in the Administrative macro-area (i.e., authorities responsible for territorial management) require differentiation according to the areas of application (Figure 9), as do the actions planned in Technical Geoconservation. Broadly speaking, geoconservation actions must be planned according to the geomorphological systems in which they are applied (since a coastal system is different from a river or a Karst hypogeum in terms of space–time evolutionary dynamics), and planned action must aim at achieving a dynamic balance between the different geo- and bio-systems, as well as anthropic systems, recognizable in the area. In the same geomorphological system, the anthropic component has a different impact on and value in the three areas of application, both in terms of the effects as a geomorphic agent and the primary human need for the "stabilization and safety of places and buildings". In protected natural contexts, such as parks or protected areas, the environment is affected in a relatively marginal way by anthropic activity compared to urban, urbanized, or anthropized contexts, where the action of man as a geomorphic agent is more evident. In some contexts, geoheritage and geodiversity can be closely connected to local sociocultural heritage; in these cases, the interventions aimed at geoconservation acquire significant additional value, as they also protect sociocultural heritage. The following text refers to the Scientific Research actions in Applied and Popularizing geoconservation for the administrative macro-area, focusing on the impact of anthropic components.

In natural and protected areas (emerged or submerged, epigeal or hypogeal), the fulcrum of planned geoconservation actions is represented by the maintenance of dynamic equilibrium between geo- and bio-systems, while the anthropic component represents an external element to the system and, as such, must adapt to the needs of the system itself (Figure 9a). Once the geo- and bio-contexts, as well as their related systems, are defined it is appropriate to treat separately, and in depth, the anthropic factor and its impact (Basic Geoconservation data collection). The anthropic impact includes the impact of both tourism (traditional, ecological, or sustainable tourism; emerged environments—epigean or hypogeum; or submerged environments) and that represented by the presence of facilities or service structures for touristic use or for the management of the natural area. The presence of human structures with historical and cultural interest (such as archaeological areas, ancient mills, trenches and war shelters, historical works of water reclamation, and stabilization of slopes) or of current interest (shelters, equipped areas, tourist routes, railways, etc.) has a tangible effect on evolution, regarding both changes already made

and possible future developments. For example, poorly maintained paths may become preferential pathways for stormwater runoff, leading to the various negative consequences of faulty runoff water regimentation; rock falls along a "via ferrata" are possible when not periodically checked and maintained; and the poor management of an anthropogenic artifact, inserted and stabilized in a PL, can have devastating effects on biodiversity. Thus, in Applied and Popularizing Geoconservation (data processing and action planning), Scientific Research macro-area actions for the Administrative macro-area are devoted to (i) the evaluation and classification of GSs and PLs; (ii) the definition of potential risks; (iii) the establishment of best practices according to the geo-context; and (iv) the planning of geoconservation actions. In Technical Geoconservation (action implementation), Scientific Research macro-area actions for the Administrative macro-area are devoted to (i) promoting the use of environmentally friendly and ecologically sustainable technologies as well as techniques; (ii) defining technical issues in suggested practices for geoconservation actions; and (iii) evaluating as well as monitoring the effect of geoconservation actions on GSs and PLs.

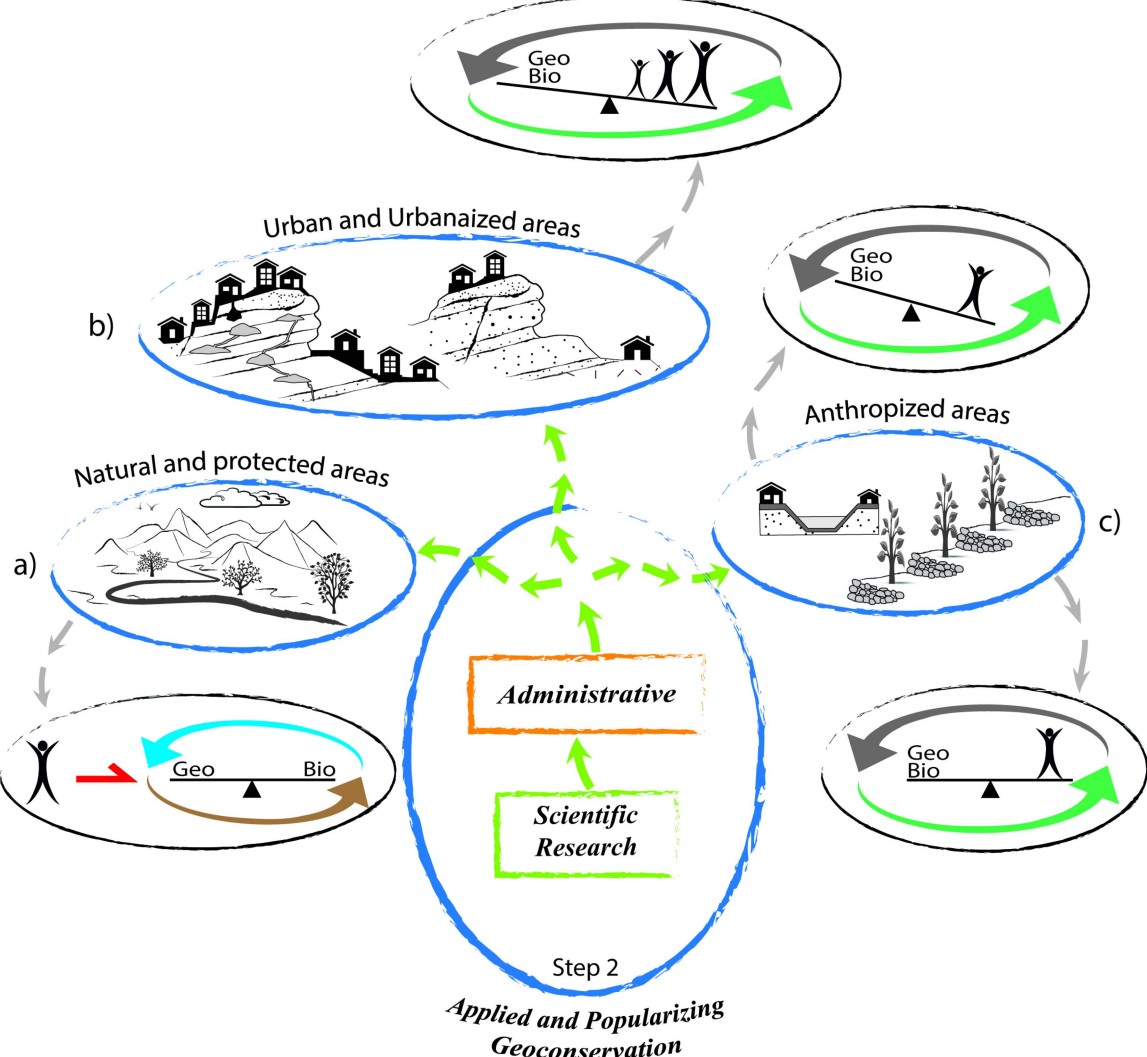

**Figure 9.** Geoconservation and areas of application. (**a**) Natural and protected areas; (**b**) urban and urbanized areas; and (**c**) anthropized areas. The Scientific Research macro area supports the Administrative macro area in all areas of application (green arrows), where the anthropic component has a different role and impact (gray arrows). Silhouette of trees and buildings designed by Freepik.

In urban and urbanized areas, geo- and bio-systems are faced with the ever-increasing impact of the anthropic component (Figure 9b). Urban geodiversity is represented by its geomorphological components [49], the presence of archeo-geosites [50,51], building stones of significant interest [52], the presence of natural hypogea, and the presence of significant outcrops—all elements that can contribute to create an attraction for urban tourism. In these areas, geoconservation actions must take into account the priority needs of the anthropic component in terms of stability and safety. The case of underground cavities (natural or artificial) in urban areas is presented as an example. Hypogea can represent a tourist attraction but also a risk when collapse phenomena occur, generating sink-holes that involve buildings on the surface. In this case, Applied and Popularizing Geoconservation actions must be dedicated to their classification according to risk, identifying those potentially more dangerous with respect to the anthropic component and planning actions devoted to geoconservation, stability, and safety (data processing and action planning). Technical Geoconservation actions must be dedicated to defining technical issues in suggested practices, both for geoconservation actions and stability as well as safety actions, and to evaluate as well as monitor the effect of the actions on GSs and PLs. With regard to the types of technology and techniques to be used, the priority represented by the implementation of safety and stabilization with respect to the anthropic factor leads to opting for techniques that are perhaps less environmentally friendly and sustainable. In this case, the topic of virtual geoconservation can be included: if a cavity requires complete or partial occlusion to ensure surface stability, a detailed 3D survey with laser scanning and image as well as video acquisition can allow it to be preserved.

In anthropized areas, human actions substantially modify the landscape, both negatively (reducing its morphological and biological diversity, increasing the possible risks) and positively (improving its stability, increasing morphological and biological diversity) (Figure 9c). In these areas, Applied and Popularizing Geoconservation actions are devoted to highlighting the risk factors and proposing actions, for the reduction thereof, to ensure balance and restore or create morphological diversity. In the case of rivers, sometimes rectified and cemented for "human needs", areas should be identified for actions towards increasing the morphological diversity (vertical and horizontal) prior to reaching inhabited centers; creating ad hoc areas (that also serve as bases for biological diversity) is included among the actions of geoconservation, preserving a river's role as an energy and mass (solid and liquid) transfer tool in its morphological system.

## 9. Conclusions

The term geoconservation is a very broad term that includes many aspects and issues related to the geo-context.

This paper focuses on a schematization of geoconservation concepts and applications, distinguishing between basic geoconservation, applied and popularizing geoconservation, and technical geoconservation. Basic geoconservation (data collecting) is devoted to the definition and characterization of GSs and PLs, analyzed within the environmental, cultural, and socioeconomic contexts in which these are included. Applied and popularizing geoconservation (data processing and action planning) is devoted to the evaluation and classification of GSs and PLs; geoheritage management database production; geoconservation general action guidelines; territorial planning guidelines; geoheritage and scientific as well as territorial knowledge dissemination; environmental issue dissemination; and geological tourism. Technical geoconservation (action implementation) is devoted to the definition and planning of short-, medium-, and long-term geoconservation actions; the definition and planning of small-, medium-, and large-spatial-scale geoconservation interventions; the evaluation, proposal, and validation of the possible use of mixed/integrated consolidation/stabilization and geoconservation techniques; and support activities for geoconservation actions carried out in non-geo-contexts.

To be effective and last over time, geoconservation actions must involve several contexts of the natural and human worlds. Several aspects identify geoconservation and

its related items (geoheritage, geodiversity, and geodissemination): scientific aspects *s.s.* and *s.l.*; technical, administrative, and political aspects; and cultural aspects. These aspects are different according to (i) the objects directly or indirectly involved in geoconservation actions; (ii) the area of application (protected and unprotected natural areas; emerged or submerged or mixed areas; and urban and/or urbanized and/or anthropized areas); (iii) the final goals; and (iv) the final end users.

Geoconservation actions are devoted to the conservation of geosites as the basic unit of the Earth's geological heritage, which also includes areas of geological interest and objects, that must be analyzed in the context of the PLs in which they are included.

Geoconservation actions must be planned according to the geomorphological systems in which are applied, and planned action must aim at achieving a dynamic balance between the different geo- and bio-systems as well as anthropic systems recognizable in the area.

Geoconservation actions, in terms of studies, techniques to be used, and implementation, are different depending on the areas of application, i.e., natural and protected areas or urban and/or urbanized and/or anthropized areas, because issues related to these areas are necessarily different, not regarding the definition and characterization of geoheritage but in their application and the definition of "anthropic impact and needs". The schemes proposed in this paper highlight (i) the fundamental and central role that geo-scientific research plays in the study as well as production of materials that can be used in other contexts, and (ii) the Scientific Research actions in Applied and Popularizing Geoconservation for the Administrative macro-area, focusing on the anthropic component impact both in terms of effects as a geomorphic agent and of primary human needs for the "stabilization and safety of places and buildings", which may "interfere" with the need to preserve the natural environment. Modern technologies allow, through dedicated and in-depth sector studies, one to reconcile the two needs, while guaranteeing stabilization/consolidation/safety and geo-preservation/fruition at the same time. The programming of consolidation, stabilization, and geoconservation interventions requires an accurate definition of (1) the geological as well as geomorphological aspects of the territory and their space–time evolution in the medium and long term; (2) the geological as well as geomorphological risk factors linked to both natural and anthropic risks; (3) the expected damages related to natural risks and man-made risks; and (4) the structural state as well as maintenance conditions of the human structures and infrastructures present.

Geoconservation actions can be both material (acting on an object or area) and virtual; virtual and augmented reality represent important tools for geoconservation actions, allowing for the creation of usable objects that otherwise could be difficult to access by end users with limited motor or sensory capacity. Geoconservation also means making an object usable, without linguistic (in terms of content and language used) or physical barriers, when possible, or choosing and suitably preparing sites/paths/contents that allow for this.

**Author Contributions:** Conceptualization, E.P., M.B. and S.I.G.; methodology, E.P., M.B. and S.I.G.; validation, E.P., M.B. and S.I.G.; formal analysis, E.P., M.B. and S.I.G.; investigation, E.P., M.B. and S.I.G.; resources, E.P., M.B. and S.I.G.; data curation, E.P., M.B. and S.I.G.; writing—original draft preparation, E.P.; writing—review and editing, E.P., M.B. and S.I.G.; visualization, E.P., M.B. and S.I.G.; supervision, E.P., M.B. and S.I.G.; project administration, E.P., M.B. and S.I.G.; funding acquisition, M.B. and S.I.G. All authors have read and agreed to the published version of the manuscript.

**Funding:** This research received no external funding.

**Institutional Review Board Statement:** Not applicable.

**Informed Consent Statement:** Not applicable.

**Data Availability Statement:** Not applicable.

**Conflicts of Interest:** The authors declare no conflict of interest.

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
