# Peer review of "Geoheritage and Geoconservation: Some Remarks and Considerations"

_sustainability, doi:10.3390/su15075823_

Round 1

Reviewer 1 Report

- I suggest to change the title, writing only: Geoconservation: some remarks and considerations. - The text is too long, sometimes repetitive and therefore difficult to read. The flowcharts of Figs. 1, 3, 4, 9, look more like video games with no real utility. I would delete them or redraw. - The concept of Geoheritage is broaded by inserting biological and anthropic aspects. Even if from my point of view this concept is wrong, the Authors should focus on this aspect. For example, there is a contradiction between the beginning of chapter 5 in which the geoheritage seems to include geosites and Physical Landscape, while in table 1 it is the PL which includes the geoheritage. Perhaps it is appropiate summarize the first 4 chapters, keep only the definitions they consider valid and focus on aspects of the various geoconservation methods. Simplyfying the text, using shorter sentences, eliminating repetitions. All to enhance what is reported on geoconservation. For these reasons, the article needs further revision.

Author Response

Please,  in the attached file the reply to the reviewer's comments.

Reviewer 2 Report

The article is devoted to a topical topic - the policy and management of geo heritage. The authors analyzed the main aspects of the formation of geodiversity, geographical specificity, and proposed specific measures to solve the problem. Graphic materials are presented in good quality, references are justified.

Author Response

(The authors gave the same response as above.)

Reviewer 3 Report

Dear authors,

I find the manuscript a too far-fetched and complex exercise from a conceptual point of view and especially as a proposal for the application of the concepts of geoconservation and geoheritage. The authors propose a complication of something that should be as simple and attractive as possible, from the point of view of its application in nature conservation and land use management. Those who come across this text might think that these themes (geoheritage, geoconservation, geodiversity) are at a level where there is no consensus in the scientific community regarding their conceptualisation and methodologies for their application. This could have been the case 30 years ago but not today, when a huge literature has already been produced, much of it with wide scientific recognition and when many initiatives and programmes are being implemented around the world. Similarly, this paper suffers from a lack of application of the ideas presented, being a mere theoretical exercise.

However, essays like this are part of the scientific discussion on these issues and should not be prevented from being published, unless they have gross conceptual and methodological errors. Therefore, I am of the opinion that this work has conditions to be presented as a publication after a major revision. 

I would like to draw attention to some aspects that should be revised.

- Complete revision of the English, as there are many passages in the text that due to the incorrect language misrepresent the content; I suggest a careful and complete revision, preferably by a specialist;  

- In many cases there is repetition of content and frameworks, as a result of the division into chapters; for example, in chapter 1 some concepts are presented, which happens again in subsequent chapters; the structure of the work should be rethought so that the first point clearly places the problems that this work intends to discuss and the following points are more specific.

- the legends of figures and tables should be revised; although, in this case, the tables and figures depend a lot on their interpretation through the reading of the main text, as a rule they should be legible by themselves, through their legend; I suggest greater clarification and objectivity of the legends.

- Some terms need revision; for example, the use of hyphens appears many times, others not (geo-heritage should be replaced by geoheritage), in other cases there are spelling mistakes (geocities).

- It would be beneficial to reduce the number of tables and figures to simplify and facilitate reading; after a certain point it is impossible to follow the text due to the complexity of elements; I suggest making fewer figures, composed with information distributed among the various figures and tables.

Author Response

(The authors gave the same response as above.)

Round 2

Reviewer 1 Report

No comment

Reviewer 3 Report

The authors have taken into account the essentials of my observations and have carefully revised the manuscript, following them as much as possible. I believe that the manuscript is now more readable and in a better condition to be published in a final version in the Sustainability journal.